A dual-phase deep learning framework for advanced phishing detection using the novel OptSHQCNN approach

Meda Srikanth 1
Srinivas Vangipuram Sesha 2
Rao Killi Chandra Bhushana 3
Ramesh Repudi 4
http://orcid.org/0000-0002-1560-8719 Yamarthi Narasimha Rao 5 y.narasimharao@vitap.ac.in
1 Department of Computer Science and Engineering, RVR&JC College of Engineering , Guntur, Andhra Pradesh , India
2 Department of Information Technology, RVR&JC College of Engineering , Guntur, Andhra Pradesh , India
3 Department of Computer Science and Engineering, Koneru Lakshmaiah Education Foundation , Vaddeswaram, Andhra Pradesh , India
4 Department of Computer Science and Engineering, KKR & KSR Institute of Technology and Sciences , Andhra Pradesh , India
5 School of Computer Science and Engineering, VIT-AP University , Guntur, Andhra Pradesh , India
Chicco Davide
Electronic publication date: 2025 Jul 17
Publication date: 2025
Volume: 11
Electronic Location ID: e3014
Received 2025 Jan 31; Accepted 2025 Jun 18
Copyright: © 2025 Meda et al.
Copyright year: 2025
Copyright holder: Meda et al.
License: This is an open access article distributed under the terms of the Creative Commons Attribution License, which permits unrestricted use, distribution, reproduction and adaptation in any medium and for any purpose provided that it is properly attributed. For attribution, the original author(s), title, publication source (PeerJ Computer Science) and either DOI or URL of the article must be cited.
License URL: https://creativecommons.org/licenses/by/4.0/

Keywords: Phishing attacks, Red kite optimization algorithm (RKOA), Convolutional block attention module (CBAM), Shallow hybrid quantum-classical convolutional neural network (SHQCNN), Optimized bidirectional encoder representations from transformers (OptBERT)

Funding: The authors received no funding for this work.

==============================
Background

Phishing attacks are now regarded as one of the most prevalent cyberattacks that often compromise the security of different communication and internet networks. Phishing websites are created with the goal of generating cyber threats in order to ascertain the user’s financial information. Fake websites are frequently created and circulated online, which results in the loss of essential user assets. Phishing websites can result in monetary loss, intellectual property theft, damage to one’s reputation, and disruption of regular business activities. Over the past decade, a number of anti-phishing tactics have been proposed to detect and reduce these attempts. They are still imprecise and ineffective, though. Deep Learning (DL), which can precisely learn the intrinsic features of the websites and recognize phishing websites, is one of the innovative techniques utilized to solve this issue.

Methods

In this study, we proposed a novel OptSHQCNN phishing detection method. Pre-deployment and post-deployment are the two phases of the proposed methodology. The dataset undergoes preprocessing in the pre-deployment phase, which includes data balancing, and handling invalid features, irrelevant features, and missing values. The convolutional block attention module (CBAM) then extracts the main characteristics from web page code and linkages. The red kite optimization algorithm (RKOA) selects the significant key attributes in the third stage. The final phase involves classifying the data using the Shallow hybrid quantum-classical convolutional neural network (SHQCNN) model. To improve the effectiveness of the classification approach, the hyperparameters present in the SHQCNN model are fine-tuned using the shuffled shepherd optimization algorithm (SSOA).

Results

In the post-deployment phase, the URL is encoded using Optimized Bidirectional Encoder Representations from Transformers (OptBERT), after which the features are extracted. The retrieved properties are fed into a trained classifier. Next, a prediction of “phishing” or “Legitimate” is produced by the classifier. With a maximum of above 99% accuracy, precision, recall, and F1-score, respectively, the investigation’s findings showed that the suggested technique performed better than other popular phishing detection methods. The creation of a security plugin for clients, browsers, and other instant messaging applications that operate on network edges, PCs, smartphones, and other personal terminals can be aided by these findings.

Introduction

Phishing, a fraudulent practice that uses technological and social strategies to get financial and personal information from unsuspecting consumers without authorization, is still a common cybercrime (Alohali et al., 2023). One of the well-known techniques is email spoofing, which is the production of false emails with a faked origin that are frequently shared on social media sites and pose as trustworthy organizations in order to fool users into visiting phoney websites and divulging private information such as usernames and passwords (Kumar, Jaya & Rajendran, 2023; Van Geest et al., 2024; Prasad & Chandra, 2024). The risk is increased when hackers utilize devices to install malicious software, making it easier for unauthorized users to access and intercept user passwords.

Email, forums, URLs, messaging applications, text messages, and phone calls are just a few of the venues that phishers utilize to get user information (Do et al., 2024). Their fraudulent content frequently imitates authentic websites, prompting people to engage and provide personal data (Vidyasri & Suresh, 2025; Zhu et al., 2024). Financial gain or theft of identities is the primary goal of phishing, which disrupts businesses all around the world (Hendaoui & Hendaoui, 2024).

One significant non-profit organization that collects, analyzes, and disseminates information on global phishing activities is the Anti-Phishing Analysis Group. With over 300,000 phishing attempts documented in July 2023, the spike in phishing attempts from 2022 to 2023 emphasizes the need for cyber security procedures (Bozkir, Dalgic & Aydos, 2023; Asiri et al., 2024; Butt et al., 2023). Webmail is still a popular target, as evidenced by the sharp rise in attempts at phishing on well-known companies from 600 to 1,100 each month, highlighting the ongoing danger (Albahadili, Akbas & Rahebi, 2024). The Anti-Phishing Act of 2016 and other strict California legislation are aimed at punishing those who commit phishing attacks with penalties of up to 5 years in prison (Yamarthy & Koteswararao, 2024).

Criminals use deceptive emails to spread false information to a large audience and fabricate illegal copies of trustworthy websites and messages, particularly those from financial institutions. This phishing technique, which uses language and emblems from reliable companies, takes visitors unexpectedly and directs them to fake websites, raising the possibility of data exploitation (Nanda & Goel, 2024). Many businesses require additional technology to identify bogus URLs, even if phishing avoidance is of utmost importance. Bayesian addition regression trees (BART) and graph convolutional networks (GCN) are two deep learning (DL) techniques that have demonstrated effectiveness in detecting features in observed datasets (Das Guptta et al., 2024).

Although blocklists are the foundation of traditional URL detection, techniques such as the domain generation algorithm (DGA) have made it harder to detect malicious URLs that are not on the list. DL techniques are the subject of recent phishing detection (PD) research, using models like XGBoost for text preparation in email bodies and URLs (Sahingoz, Buber & Kugu, 2024; Roy, Kumar & Singh, 2024; Alsubaei, Almazroi & Ayub, 2024). Through the examination of email frameworks, links to websites, files, information of the sender, and metadata, extreme gradient boosting (XGBoost) effectively processes massive databases, managing noise and extracting key elements needed for effective phishing classification.

Modern processes require a shift from conventional approaches and a greater level of human involvement. Graph convolutional networks (GCN), recurrent neural networks (RNN), and artificial neural networks (ANN) are examples of modelling approaches that use robust chronological datasets to improve the effectiveness of phishing detection tools significantly (Karim et al., 2023; Shirazi et al., 2023). These neural network models are essential in the field of digital forensics because they help detect and stop phishing attempts, making the internet a safer place.

Motivation and our research contributions

Phishing attacks pose a significant threat to internet users, resulting in financial losses, data breaches, and reputational damage. Despite the increasing complexity of these attacks, many detection systems struggle to accurately identify phishing websites due to outdated methods that cannot keep up with evolving online threats. This study aims to address these challenges by using advanced deep learning techniques, such as OptSHQCNN and optimized feature extraction, to improve phishing detection accuracy. By integrating state-of-the-art algorithms and reliable datasets, the research seeks to enhance the efficiency and scalability of phishing detection systems, mitigating the risks associated with these cyber threats.

The major key contributions of this research are as follows, To extract primary features, the convolutional block attention module (CBAM) is utilized to efficiently capture significant characteristics from URLs and webpage code for accurate phishing detection.

The red kite optimization algorithm (RKOA) is used to select the most essential features, minimizing the feature set and improving the computational efficiency of the model.

To classify phishing websites, a novel shallow hybrid quantum-classical convolutional neural network (SHQCNN) approach is introduced, leveraging their strengths for enhanced detection capabilities.

The shuffled shepherd optimization algorithm (SSOA) is utilized to fine-tune the parameters presented in the SHQCNN approach.

For URL encoding in the post-deployment phase, Optimized Bidirectional Encoder Representations from Transformers (OptBERT) is utilized to facilitate high-quality feature extraction that supports accurate predictions.

Organization of this research: This study is structured as follows. The relevant literature and the identified research gap are discussed in “Review of Existing Related Works”. “Overview of Proposed Methodology” explains the proposed multiple classes URL-classification method. The experimental design, findings, and comparative evaluation are covered in “Result and Discussions”. The work is finally concluded and possible future research areas are highlighted in “Conclusion and Future Scope”.

Review of existing related works

Since phishing site identification is essential for stopping cyberattacks, a lot of work was done to develop a dependable and efficient solution. But even with the advancement of numerous techniques, the ideal system is still too far off. We will concentrate on deep learning-based methods in this work because of the significant results that can be obtained with them.

In order to detect phishing URLs, Ozcan et al. (2023) introduced two hybrid deep learning-based approaches, deep neural network long short-term memory (DNN-LSTM) and deep neural network bilateral long short-term memory (DNN-BiLSTM) models, which make use of both character embedding-based features and manually created NLP features. The suggested models can identify strong character relationships while incorporating the qualities of advanced NLP characteristics.

CNN-Fusion is a lightweight and efficient phishing URL detection technique presented by Hussain et al. (2023). Our main concept is to obtain features at multiple levels by simultaneously deploying many one-layer convolutional neural network (CNN) variants with different kernel sizes. Given the possibility of a significant spatial correlation between phishing and benign URL changes, they select SpatialDropout1D, which strengthens the model and keeps it from learning the training data.

Binary Bat was suggested by Kumar, Jaya & Rajendran (2023) to detect phishing URLs on web pages. Neural networks that classify network URL webpages using a classification-like technique are designed using the binary bat algorithm. It was employed for the first time in this field, and it was essential to the forefront of the field of study.

Zhu et al. (2024) provide a phishing identification model called phishing detection based on hybrid features (PDHF), which combines optimal artificial and automatic DL characteristics. The newly created feature significance assessment index and an enhanced bidirectional search algorithm are used to eliminate redundant attributes in order to achieve the finest fake phishing characteristics. Using a chaotic quantized attention mechanism and a one-dimensional characters CNN, deep features are extracted from URLs in order to increase the effective duration of detecting phishing.

A hybrid deep neural network-based method called LBPS was developed by Wen et al. (2023) to identify phishing scam accounts. Its efficacy is confirmed on Ethereum. The LBPS model offers a new method for analyzing transaction records, using the long short term memory fully convolutional neural network (LSTM-FCN) to capture the temporal characteristic from all transaction records of a target account and the BP neural network to determine the implicit connection among characteristics obtained from transaction records.

Sahingoz, Buber & Kugu (2024) presented the development of a deep learning-based phishing detection system that uses five distinct algorithms: attention networks, recurrent neural networks, convolutional neural networks, artificial neural networks, and bidirectional recurrent neural networks. Fast web page categorization utilizing URLs was the system’s primary goal. The system’s efficiency was evaluated by gathering and disseminating a comparatively large dataset of labelled URLs, which included roughly five million records.

In order to increase FS in ML models for the detection of phishing websites, Shafin (2024) suggested an approach that makes use of eXplainable AI (XAI). In particular, we use aggregated local interpretable model-agnostic explanations (LIME) to identify particular localized patterns and Shapley Additive exPlanations (SHAP) for global perspective. By identifying the most useful traits, the suggested SHAP and LIME-aggregated feature selection (SLA-FS) framework makes phishing detection more accurate, quick, and flexible. We test the efficacy of this method by comparing the efficiency of three ML models before and after FS on a current web phishing dataset.

An EGSO-CNN model was suggested by Barik, Misra & Mohan (2025) to detect web phishing by optimizing deep learning (DL) techniques and incorporating features. An additional set has been developed to tackle the availability of current updated phishing datasets. Preprocessing and extraction of features are done using the StandardScaler and variational autoencoders (VAE). The model’s performance is optimized using the enhanced grid search optimization (EGSO) technique.

To identify phishing attempts, Albahadili, Akbas & Rahebi (2024) presented a hybrid approach that combines swarm intelligence, machine learning, and deep learning techniques. Deep learning based on the GAN balances the dataset in the initial phase. The convolutional neural network then extracts the main characteristics from the web page’s code and links in the second step. The white shark optimizer algorithm chooses the key properties in the third step. The LSTM neural network classifies the samples in the final stage.

Nanda & Goel (2024) present a brand-new method for identifying phishing URLs: the bidirectional long short-term memory gated highway attention block convolutional neural network (BiLSTM-GHA-CNN). Whereas the CNN extracts prominent information, the BiLSTM records contextual features. The BiLSTM-CNN architecture’s use of the highway network allows for the quick convergence and capture of essential features. Additionally, the CNN and BiLSTM output characteristics are weighed using a gating method. A summary of related work is shown in Table 1.

Table 1 Summary of related works.

Reference	Method used	Advantage	Disadvantage	
Ozcan et al. (2023)	DNN–LSTM	Combines high-level NLP features with character-based connections for better phishing URL detection	Computational complexity due to hybrid model integration	
Hussain et al. (2023)	CNN	Lightweight, robust against overfitting, and effective feature selection	Limited scalability due to dependency on specific kernel size variations	
Kumar, Jaya & Rajendran (2023)	Binary bat algorithm	First-time application in phishing detection; improves classification performance.	Relatively untested approach in diverse phishing detection scenarios	
Zhu et al. (2024)	Hybrid features (PDHF)	Extended phishing detection time and optimized feature selection for better accuracy	Potential overhead from combining multiple feature optimization processes	
Wen et al. (2023)	LBPS model using BP neural	Novel transaction record analysis captures temporal and implicit relationships effectively.	Domain-specific applicability limits broader usage.	

Research gap

The growing complexity of phishing attempts is threatening cybersecurity. Phishing is an increasing threat to people and companies alike. It is an approach that blends technology with social engineering. Phishing primarily attempts to obtain sensitive information, including financial or personal information, secretly, putting a person’s privacy and economic stability at risk. One common hacker technique is email spoofing, which involves sending bogus emails with fake sender addresses. In an effort to deceive recipients into clicking on bogus links or disclosing personal information, these emails usually pretend to be from trustworthy sources. Phishing attempts that are shared on social media platforms have a more significant effect and reach.

Identity theft is one problem that might arise from this, in addition to financial losses. Despite defences, phishing attempts are becoming more frequent. Traditional systems, especially those that rely on static blocklists, are unable to keep up with the ever-evolving tactics of attackers. Since advanced tactics like domain generation algorithms (DGA) make detection even more challenging, new approaches to anti-phishing activities are needed. Current methods frequently focus on specific aspects of phishing, including text analysis in emails or URLs. There is a severe shortage of comprehensive models that consider the various elements of phishing schemes, including attachments, URLs, sender information, images, and textual content.

Overview of proposed methodology

The two-stage deep learning approach is used in the proposed phishing detection methodology. The dataset is preprocessed during the pre-deployment stage, which includes cleaning and balancing, and then features are extracted using the CBAM model. The most pertinent features are chosen by RKOA and utilized to train a classifier.

Optimized Bidirectional Encoder Representations from Transformers (OptBERT) are used to encode URLs in the post-deployment phase, which improves the feature extraction procedure. To ascertain if the URL is legitimate or phishing, the classifier makes use of the information that was extracted. The approach combines optimization methods with deep learning models to produce superior performances. The overall framework of the proposed methodology is shown in Fig. 1.

Figure 1 Overall framework of the proposed methodology.

Preprocessing

To ensure that the model is trained on high-quality data, we begin with data preprocessing, which includes handling invalid and irrelevant features, as well as dealing with missing values. Additionally, the class imbalance issue, where phishing URLs are underrepresented, is tackled using the auxiliary classifier generative adversarial network (ACGAN). ACGAN generates synthetic phishing URLs, enhancing the minority class and improving model robustness. During this phase, the following issues are addressed: Dealing with “invalid” features: The feature qty_questionmark_domain, which shows how many question marks are in the name of the domain, is one instance of a feature that is considered invalid. An underscore, comma, semicolon, or question mark are among the symbols prohibited in domain names under the DNS RFC. The dataset had 16 incorrect characteristics.

Handling ‘irrelevant’ features: One property that may be easily statistically deduced from other features is the qty_hyphen_url characteristic, which displays the number of hyphens (-) symbols. This feature is legitimate, but four more features count how many hyphens are in the domain, folder, file, and parameter. This feature is, therefore, unnecessary. After closely examining the features and applying our domain knowledge, we identified and eliminated 19 unnecessary characteristics.

Dealing with cases where “missing values” occur: Some values were absent from specific instances.

Data imbalance

Using the auxiliary classifier generative adversarial network (ACGAN), we addressed the problem of class imbalance in the phishing dataset, where legitimate URLs significantly outnumber phishing ones. Though they were first taken into consideration, conventional rebalancing approaches like random oversampling and SMOTE either added redundancy or were unable to recognize the intricate patterns present in phishing URLs. ACGAN, on the other hand, provides a more advanced approach by producing superior synthetic samples for the minority class (phishing URLs). The generator and the discriminator are its two primary parts. The generator learns to create synthetic phishing URLs conditioned on class labels, while the discriminator classifies both real and fake samples, assigning class labels. This conditional structure allows the generator to produce diverse, semantically meaningful synthetic data, enhancing the representation of phishing URLs in the dataset.

In an adversarial setting, the generator and discriminator are updated alternately during the training phase. In particular, the discriminator separates created samples from real data and provides input to help the generator improve while the generator creates synthetic samples from random noise. ACGAN enhances the model’s capacity to identify phishing websites by maintaining the structural patterns common to phishing URLs, in contrast to conventional techniques. With a batch size of Z and a learning rate of Y, the model was trained across 200 epochs. We evaluated using standard classification metrics, paying particular attention to F1-score, precision, and recall for the minority phishing class. By lowering overfitting and boosting the model’s resilience, the use of ACGAN significantly increased recall and F1-score for phishing URLs, increasing the model’s efficacy in phishing detection. To balance the dataset, we used the self-paced ensemble and auxiliary classifier generative adversarial network (SPE-ACGAN) method. This section initially introduces every module’s operating concept and the overall framework of SPE-ACGAN as proposed in this article. Details of the algorithm’s implementation and the datasets utilized in the proposed algorithm are then provided.

SPE-ACGAN

We reproduced the training instances from two dimensions in order to account for the imbalance of the network traffic training samples in NIDS. The majority class’s sample size is reduced by utilizing SPE, while the minority class’s sample size is increased by employing ACGAN.

SPE

The fundamental idea behind SPE, a structure for unbalanced categorization, is to provide a concept of categorization hardness and coordinate data difficulty by creating an original undersampled dataset using self-paced undersampling. Figure 2 depicts the SPE procedure. The input, the unbalanced dataset, is first split into a majority set (N) and a minority set (P) by SPE. Subsequently, every instance in N is sorted at random into k bins based on its categorical hardness; every group has a total categorized hardness. To finish the resampling, the procedures above are repeated until the number of majority class samples matches the number of minority class instances or reaches the predetermined number.

Figure 2 SPE framework.

SPE maintains the overall category difficulty of each bin constant in order to produce a balanced dataset. A “categorical hardness function,” such as Absolute Error, MSE, and Cross Entropy, is used to determine the hardness value. Equation (1) provides the categorical hardness of this specimen (x) for each model:

(1) Hx=H(x,y,F).

While Bl represents the grade of hardness of the lth box, Eq. (2) provides the hardness grade:

(2) Bl={(x,y)|l−1k≤Hx=H(x,y,F)≤lk}H(⋅)∈(0,1).

ACGAN

Figure 3 illustrates the structure of ACGAN, which is primarily made up of Generation (G) and Discrimination (D). This is how ACGAN operates: The network creates an arbitrary set of noise values z in order for ACGAN to function.

Figure 3 Architecture diagram of ACGAN.

The discriminator D trained by Xren determines whether the generated Xfake is actual data, and if it is virtual data, what is the likelihood of belonging to every category, accordingly. The generator G converts the noise data z into Xfake of the appropriate group based on the input of the specified category. The loss function determines the error. The parameters are to be updated by the generator G.

Overall model architecture

The two steps of the SPE-ACGAN resampling approach are carried out by SPE and ACGAN, correspondingly. The minority class samples are first oversampled and then fed into ACGAN to boost the amount of minority class data. The majority class samples are then undersampled and fed into SPE to decrease the amount of majority class instances. Figure 4 illustrates the process of the SPE-ACGAN resampling approach.

Figure 4 The overall framework of SPE-ACGAN.

Feature selection

By concentrating on the most essential data, choosing pertinent features improves DL model accuracy and lowers computing complexity. By removing redundant or unnecessary features, effective feature selection enhances model interpretability and defends against overfitting. For the best feature selection in this position, the RKOA can be applied. RKOA is a novel meta-heuristic method that mimics the social life of red kites (RKs). Typically, the RKs build their nests close to areas with lakes and woods that are suitable for hunting. In order to avoid being stuck in the local optimum, the meta-heuristic approach must first traverse the issue of exploring space fit. It then employs the best performance from the last iterations as it gradually progresses from the exploration to the exploitation stages. RKOA follows three crucial stages that are specified:

The first stage is where birds are primarily found: According to Eq. (3), the RK’s position is first set arbitrarily during this stage as:

(3) Posi,j(t)=lb+rand×(ub−lb),i=1,2,….,nandj=1,2,…,d.

In this case, n indicates the size of the population, d indicates the problem’s dimensions, and rand represents the randomized integer between zero and one. Posi,j(t) is the ith location of the RKs at iteration t, while lb and ub define the lower and upper limits, respectively.

The selection of the phase of the second leader: Eq. (4) is used to determine the leader:

(4) Best(t)→=Posi(t)→iffi(t)<fbest(t)

wherein the spot of the optimal bird from iteration t is represented by Best(t)→. Posi(t)→ gives the spot of ith RK from the iteration t, fi(t) shows the values of the bird estimation function during the iteration t, and best(t) shows the value of the assessing function of the best birds from the iteration.

The third stage: By considering a lowering co-efficient ( D) according to Eq. (5), it can be inferred that RKs must proceed gradually from the exploration to the exploitation stages.

(5) D=(exp⁡(tt−max()tt−max)−10).

The maximal iteration is denoted by t−max, where t represents the current iteration. The birds use Eqs. (6) and (7) to improve their positions:

(6) Posinew(t+1)→=Posi(t)→+Posmi(t+1)→

(7) Pmi(t+1)→=D(t)×Pmi(t)→+SC(t)→Θ(Posrws(t)→−Posi(t)→)+UC(t)→Θ(Best(t)→−Posi(t)→).

The novel position of birds is indicated by Posinew(t+1)→. It is then necessary to confirm the boundaries of the searching space in the upgrade position; this could be done by using Eq. (8),

(8) Posinew(t+1)→=max(min(Posinew(t+1)→+ub),lb).

After improving the estimation function, a new temporary position is swapped. They represent the voice of all bird’s unity and danger, and they may be achieved using the relationship below:

{SC(t+1)→=r1→UC(t+1)→=r2→ifrand≤0.5

(9) {SC(t+1)→=r3→UC(t+1)→=r1→ifOtherwise.

In the RKOA, the neighbour position is chosen at random by employing a roulette wheel based on the current positions of all the birds, and the best solution is still being discovered. According to its position and randomly selected neighbour, the RK explores new spaces as it advances based on a single factor. The RKOA approach presents weighted recognition of all the objective relevance by combining the objectives into a single objective equation. In this instance, an FF is being implemented to combine both FS aims, as shown in Eq. (10). The selected features are shown in Table 2.

(10) Fitness(X)=α⋅E(X)+β∗(1−|R||N|).

Table 2 Selected features.

Feature name	Description	
URL length	Measures the total number of characters in the URL; longer URLs may indicate phishing attempts.	
Domain age	Represents the age of the domain in days since registration; recently created domains are often malicious.	
Presence of HTTPS	Indicates whether the URL uses HTTPS for secure communication; phishing sites often lack HTTPS.	
Abnormal URL	Check if the URL contains unusual patterns, such as uncommon domain names or obfuscation techniques.	
IP address usage	It detects whether the URL directly includes an IP address instead of a domain name, which is often used in phishing.	
Special characters count	Counts the number of special characters (e.g., @, %, −), as phishing URLs often contain many.	
Shortened URL	Identifies if the URL uses a shortening service (e.g., bit.ly); shortened URLs are commonly used in phishing.	
Subdomain count	Counts the number of subdomains in the URL; phishing URLs may use excessive subdomains to mimic legitimacy.	
Domain entropy	Measures randomness in the domain name; high entropy may indicate a malicious domain.	
Top-Level domain (TLD)	Analyzes the TLD (e.g., .com, .net); malicious actors more commonly use some TLDs.	
Anchor tag URL match	Checks if anchor tags in the webpage match the primary domain of the URL; mismatches indicate phishing.	
Redirect count	Determines the number of redirections in the URL; excessive redirects are common in phishing sites.	

Phishing classification

A state-of-the-art method that combines the advantages of classical and quantum computing is the shallow hybrid quantum-classical convolutional neural network (SHQCNN), which is used for phishing classification and detection. Using classical layers for robust learning and quantum principles to handle intricate data patterns, SHQCNN facilitates effective feature extraction and classification. This hybrid approach is very effective in mitigating online threats because it improves detection accuracy, scalability, and adaptability to changing phishing strategies.

Quantum logic gates in phishing classification

The hybrid quantum-classical CNN is employed for classification, where the quantum component is utilized for feature encoding and data transformation, while the classical CNN handles the final classification. We can take advantage of both the demonstrated effectiveness of CNNs in image-based classification tasks and the benefits of quantum computing in processing high-dimensional data due to this combination. Although quantum networks are usually used in situations where quantum relationships are present in the data, we presented a hybrid quantum-classical CNN model for URL classification in this work. However, this method offers a promising technique to improve the model’s processing capabilities for high-dimensional, complex URL data, which is why we chose it. Even though traditional CNNs are excellent at identifying patterns, they might not be able to comprehend the complex linkages seen in phishing URLs properly. Quantum computing, leveraging quantum superposition and entanglement, can potentially improve the model’s ability to represent these complex features, offering a novel advantage in feature extraction and classification tasks.

A real quantum simulation was not carried out in our experimental setup. Instead, a hybrid architecture was employed to construct the quantum component, which was utilized for both data transformation and feature encoding. To make the method is computationally viable, the quantum component was simulated using traditional computer frameworks. With the help of the quantum-processed characteristics, the classical CNN then managed the final classification. This hybrid method increases the accuracy of phishing detection and improves model generalization, particularly in adversarial situations. This technique shows how quantum methods can enhance cybersecurity, especially for complex tasks like phishing URL identification, even though they are still in the early stages of development. A key component of quantum computing is quantum logic gates, which allow for the extraction of quantum information by altering the amplitude values of qubits. These gates, which facilitate operations like swapping and changing quantum states, include the Hadamard gate and the controlled-NOT gate. Like classical circuits, quantum circuits, which are made up of gates and qubits, carry out operations from left to right.

By incorporating quantum gates for optimal data processing, quantum circuits can improve computing efficiency in phishing classification. The method lowers complexity while increasing performance by building circuits with many measurement units and connecting quantum outputs to classical systems.

Variable quantum circuit

The symmetry and localization of variable quantum circuits (VQCs) allow for efficient construction by encoding input states into quantum actions. VQCs are designed to minimize parameters and maximize expressive capacity in the era of noisy intermediate-scale quantum (NISQ).

Despite being linear by nature, VQCs use entanglement or coupling to simulate the nonlinear operation of artificial neural networks. This allows for noise reduction and improves performance in multiple categorization tasks. VQCs can enhance data representation and reduce noise in phishing categorization, enabling more precise detection and classification.

Proposed SHQCNN

Quantum algorithm

The implementation of quantum neural networks typically involves three stages of the quantum algorithm: gradient computation, evaluation circuit model, and quantum data coding. The quantum algorithm’s specifics are as follows:

Encoding of quantum data. Given a standard dataset, where the appropriate label and the input vector. First, the vector element (0, π/2) is rescaled to a range of the second step, which is to translate every vector x to the N qubit product state.

(11) x→|ψ(x)⟩=⊗i=1NRy(2θj)|0⟩

The circuit model. The quantum circuit reaches the ultimate state. The traditional optimizer. In this study, the mean square error between the class label and the predicted value is minimized.

(12) J(θ)=1D∑d=1D⁡(Mθ(ψ(xd)−yd))2

While D represents the number of training datasets, it is the matching class label and ψ(x) provides the input data. When developing models for categorization, existing general quantum algorithms outperform conventional algorithms in terms of speed and accuracy by utilizing the potential benefits of quantum theoretical mathematics. The creation of all-encompassing quantum computing is still a long way off, though. We then introduce our suggested VQC, a viable method for resolving multi-classification issues.

Variable quantum algorithm

Through the effective training of mini-batch gradient descent, entanglement gates, and kernel encoding, the variable quantum algorithm (VQA) improves multiple categorizations. It reduces complexity by using shallow circuits with parameterized unitary operators and modifies quantum outputs for CNN-like hybrid models. Since VQA can manage spatial complexity and minimize noise, it can be used for the detection of phishing, allowing for faster training and more accurate classification of multiple classes. VQA was first used for envisioning classification.

In the first section ψ(x) is a mapping to the feature vector’s unit length and consists of two conjugate operators together with their inverse operator. Employing kernel encoding, Uφ(x′) produces quantum datasets ψ(x)⟩=⊗i=1NRy. An entanglement gate and two random single qubit spins make up 2θj the second section. Thirdly, the shallow quantum circuit is realized by arranging many samples in the readout layer.

To improve model accuracy, the neural network layer in the suggested VQA uncovers hidden characteristics. To maximize efficiency, model parameters are iteratively upgraded by employing a mini-batch gradient descent approach. This VQA combines kernel encoding and several output methods, allowing for effective multi-class classification, in contrast to other Variational quantum algorithms that were restricted to categorized binary data. By increasing accuracy and decreasing running time, it enhances phishing detection and provides a notable benefit over previous techniques.

Implementation and achievements

By combining quantum and classical layers, the proposed strategy provides flexibility in the placement of quantum circuits at different phases. In order to efficiently implement quantum algorithms, data must be converted into a quantum state for kernel encoding. In contrast to conventional techniques that necessitate O(N) space complexity, the SHQCNN projects data onto higher-dimensional feature spaces via kernel coding, which improves dataset distinguishability. By lowering computational complexity and increasing feature separability, this preprocessing technique improves phishing classification. (1) We determined the initial dataset’s mean value and variance. (13) u=1D∑d=1D⁡Xd

(14) σ=1D∑d=1D⁡(Xd−u).

The zero-mean normalization is applied to them. It has zero unit variance and zero mean. It will follow the typical distribution in this manner.

(2) To lessen the impact of edge strength, we normalized eigenvectors to the unit length φ(x′).

(3) Equation (15) is the product state of N′=⌈(N/2)⌉ qubits to which we mapped vectors x′. (15) x′→Uφ(x′)|0⟩⊗N′=⊗j=1N′[cos(θj)sin(θj)]=⊗j=1N′Ry(2θj)|0⟩.

The kernel coding approach, in summary, employs a feature map x′→Uφ(x′) into a vector space of greater dimension. Since N classical data may be encoded using this encoding approach using approximately ⌈(N/2)⌉ qubits, the spatial complexity is O(N/2).

Variable quantum circuit

Inspired by previous developments, the variable quantum circuit (VQC) minimizes circuit depth while approximating nonlinear functions for NISQ processors. The generator in the SHQCNN model consists of VQC, which is made up of entanglement layers and rotation layers. The circuit’s central component, these gates, enable qubit rotations and interactions. Through the use of VQC, the model effectively encodes and analyzes data, improving its capacity to classify phishing attempts.

(16) RY(θ)=(cos⁡(θ/2)sin⁡(θ/2)−isin(θ/2)(θ/2))

(17) RZ(θ)=(e−iθ/200eiθ/2)

VQC uses RY and RZ gates to rotate quantum states on the Bloch sphere, changing the phases and probability amplitudes accordingly. VQC, which forms the basis of SHQCNN, approximates goal functions and encodes input states. In contrast to deep circuits, VQC simplifies implementation by utilizing symmetry and locality from real-world datasets. This design preserves the computing economy while improving accuracy in multi-class situations such as phishing categorization.

Multi-output and optimization in SHQCNN

The proposed paradigm scales circuits to lower depth and parameters while introducing a quantum multi-output layer to handle high-dimensional input qubits effectively. This layer maintains the integrity of every characteristic while fusing data and performing parallel measurements. To increase classification accuracy and extract hidden characteristics, these results are fed into the CNN. SSOA is used to optimize the model, striking a balance between the speed of stochastic approaches and the stability of classical gradient descent. This ensures effective and reliable phishing categorization training.

Theoretical and feasibility analysis of SHQCNN

To efficiently extract and classify information, the SHQCNN combines three fully connected layers with a quantum convolutional layer. A ReLU activation layer adds nonlinearity to improve the feature mapping procedure, while the convolutional layer uses several weight matrices for comprehensive and varied extraction of features. By using dynamically shifting weight matrices, SHQCNN improves generalization and reduces overfitting, in contrast to classic CNNs that are prone to overfitting because of fixed weight progression. In addition, contrasted with conventional models, the quantum convolution layer, when implemented with VQC, allows for more accuracy, fewer layers, and lower complexity. This structure ensures robust classification performance and effective feature extraction.

Hyperparameter tuning using the shuffled shepherd optimization algorithm

In the proposed classification model, the optimal value for the hyperparameter is chosen using the SSOA strategy. By effectively identifying the ideal parameter configurations, the SSOA for hyperparameter tuning ensures optimal performance. In phishing detection tasks, SSOA achieves better accuracy, faster convergence, and less overfitting by striking a balance between exploration and exploitation.

Shuffled shepherd optimization procedure

The hyperparameters of the shallow hybrid quantum-classical convolutional neural network (SHQCNN) are fine-tuned using the SSOA, which enhances the model’s performance by finding the optimal hyperparameter set. The main goal of this part is to increase the application of the SSOA meta-heuristic approach. In SSOA, a “sheep” is any potential multivariate solution. The values of the aim function are used to group the sheep. Some sheep are called “shepherds,” and the sheep with the best goal function are called “horses” in the herd. For this reason, every shepherd may own a certain number of sheep and horses. In an attempt to herd the sheep toward the horse, the shepherd changes positions by leaping onto either the horse or one of the sheep. There are two reasons for this: (i) changing to a different, possibly better member encourages investigation, and (ii) changing to a various, possibly worse member leads to exploitation. A hierarchy is created in the algorithm when the shepherd’s site is modified, and the new objective function is not worse than the old one.

A summary of the SSOA procedure is provided below. (1) SSOA parameters are set to α,β,β max, itermax, h, and s. Where ‘iter max’ is the maximum number of permitted iterations, ‘h’ is the overall amount of flocks, and ‘s’ is the total sum of sheep.

(2) The starting point in m-dimensional search space is determined by Eq. (4). (18) Xi0=Xmin+rand∘(Xmax−Xmin),i=1,2,⋯,n(4).

While Xi0 represents the ith sheep’s initial key vector, Xmax as well as Xmin the boundaries of the design space, rand is a vector with components that are between (0, 1), the total of its machinery equals the sum of its design variables, and n denotes the sum of sheep (n = h s). As well as the multiplier “>” that operates element by element.

(3) After determining the objective function of every sheep, the flock is ordered from highest to lowest. To expand your herd, move the sheep around. One sheep is in each flock, and the first sheep are selected at random and dispersed among the flocks. Next, assemble the second set of h sheep. Whenever every flock of sheep has a chosen leader, this process is repeated.

(4) In a flock, identify every sheep. The best of the herd are called horses, while the selected sheep are called shepherds. Select a sheep flock and a horse at random; each shepherd’s step size can be determined by (19) Stepsizei=β×rand∘(Xd−Xi)+a×rand∘(Xj−Xi).

Assuming that Xi, and Xj are in an m-dimensional search space, rand is a random vector whose elements are in the interval (0, 1), and the number of procedures is determined from the total number of the aspects of the solution vectors; they are calculated using Eqs. (20) and (21). (20) α=α∘−α∘itermax×iteration

(21) β=β∘+βmax−β∘itermax×iteration.

The step size’s initial phase is zero since the first sheep in a herd are unable to have a greater offspring than itself, and the size is also zero since the final sheep in the herd cannot have a worse offspring than itself.

(5) Using Eq. (22), every sheep’s temple is determined: (22) Xitemple=Xiold+stepsizei.

While the sheep’s location is altered due to an ancient objective function. However, the shepherd’s position remains unchanged. For communication reasons, the flocks could be merged once every sheep’s location has been updated.

The optimization method repeats step 3 as many times as permitted, which is the termination criterion. Using the SSOA, hyperparameter adjustment was done to maximize the performance of the proposed SHQCNN model. In order to ensure stable convergence and avoid overfitting, a well-defined and constrained search space was created for the essential hyperparameters. In particular, the number of filters per convolutional layer varied from 16 to 128; batch sizes were chosen from discrete values {16, 32, and 64}; dropout rates were varied from 0.1 to 0.5; and kernel sizes were selected from {3, 5, 7}. The learning rate was investigated within the continuous range of 0.0001 to 0.01. Every hyperparameter was encoded according to its type: discrete and categorical parameters were represented as index-based integers and subsequently decoded during evaluation, whilst continuous parameters were normalized to the [0, 1] range. To cut down on pointless computation, the optimizer was run for 200 iterations with early stopping enabled. This precise setup greatly enhanced the suggested phishing detection model’s increased effectiveness and resilience.

Cloud-based deployment (post deployment)

BERT for URL encoding in phishing detection

Phishing detection can be aided by the abundance of information found in URLs. Feature extraction methods that concentrate on a URL’s structural elements, like domain names, path segments, and query parameters, are frequently used in traditional approaches to URL analysis. The semantic linkages between a URL’s constituent parts, which are essential for differentiating malicious URLs from lawful ones, might not be fully captured by these techniques. The advanced technique provided by Bidirectional Encoder Representations from Transformers (BERT) involves encoding the URL into a dense vector representation, which captures syntactic and semantic information from the complete token sequence.

In order to analyze the URL in both directions and comprehend the context from both the left and the right of each token, BERT employs a Transformer-based architecture. Because of its capacity to record context, BERT is a perfect fit for encoding URLs since the meaning of each token might change based on the context in which it is used. A domain like “amazon.com” could be harmless in one situation but harmful in another, for instance, if it is slightly altered (for example, “arnazon.com”). This kind of URL encoding allows BERT to spot minute patterns in token interactions that might point to phishing efforts.

Fine-tuning BERT for URL encoding

While BERT is a generic encoder, its performance is improved by tailoring the model for phishing URL identification using annotated datasets of real and phishing URLs. This technique enables the pre-trained BERT model to understand the distinct characteristics that define fraudulent URLs by adapting it to the phishing detection task. BERT’s parameters are modified during fine-tuning in response to the phishing dataset, which allows it to modify its contextualization and tokenization capabilities in order to identify phishing efforts more effectively.

While keeping the rich, contextual embeddings that BERT generates, fine-tuning entails altering the last layers of the BERT model to provide the intended outcomes regardless of whether a URL is malicious or benign. The program can learn to link specific patterns, such as dubious domain names or unusual URL frameworks, with phishing activity by being trained using a dataset of phishing URLs. The outcome is a reliable model that does an admirable task of spotting phishing URLs in a practical setting.

Benefits of using BERT for URL encoding in phishing detection

Contextual understanding: Unlike standard models, BERT takes into account the contextual interactions between URL components. For example, while the domain “google.com” might not be harmful, variations such as “gooogle.com” might be signs of a phishing attempt.

Managing ambiguities: Phishing URLs frequently use minute changes, including misspelled domain names or character replacements. Although it is bidirectional, BERT is able to recognize these details and efficiently identify problematic patterns.

Enhanced precision: Higher accuracy and fewer false positives in phishing URL identification can be achieved by fine-tuning BERT on a particular phishing detection dataset. This helps the model to understand the complex patterns unique to phishing attempts.

Feature extraction using CBAM

Lexical patterns, domain attributes, and embedded character sequences are among the syntactic and semantic aspects that the system collected and found to be pertinent to phishing identification. This created a structured representation of unstructured URL data that might be used for learning. We employ the CBAM to extract pertinent features from URLs. This module concentrates on the most crucial elements of the URL structure and web page code, improving the model’s capacity to differentiate between authentic and phishing websites. This module lessens the influence of unimportant features while allowing the network to focus on the most important ones. CBAM is a powerful yet computationally effective method that selectively focuses on pertinent information in both spatial and channel dimensions to improve feature extraction. The spatial attention module (SAM), which finds the spatial regions of interest by pooling along the channel axis, and the channel attention module (CAM), which highlights the significance of particular feature channels by examining global max and average pooling operations, are integrated sequentially.

These modules work together to give CBAM the ability to identify “what” information is essential and “where” it is. CBAM iteratively applies attention techniques to the feature maps, minimizing redundant or irrelevant features. Because of this feature, CBAM is a valuable tool for phishing detection, where it is necessary to pay close attention to essential patterns in the data’s structure and content in order to differentiate between authentic and fraudulent URLs. The detection performance is greatly enhanced by the combination of spatial and channel attention, which guarantees reliable and effective feature extraction.

Following classifier testing, the model was stored using the joblib Python package. The deployment web service that would generate predictions using the saved model was then constructed. An overview of the server-side operation following deployment is displayed in Fig. 5.

Figure 5 Overview of server-side operation.

As illustrated in Fig. 5, the client initiates the process by submitting an HTTP request to the web service that is currently operating. The format of the request is “http://serveraddress/?url=URL-to-inspect”, where URL-to-inspect is the URL that the system needs to check in order to mark it as “phishing” or “benign.” URI encoding must be used for all URLs sent to the server. The URL is taken out of the HTTP request and parsed once it reaches the web service. The URL is divided into the domain name, folder, file name, and other arguments throughout the parsing process. The first “/” indicates the end of the name of the domain after the protocol name (HTTP:// or HTTPS://) has been removed, and the last “/” indicates the beginning of the file name (if any), and any text that appears in between (if any) is regarded as the folder name. After that, two extracting feature units get the parsed URL. Features to be taken from the URL itself, like qty_dot_domain, are extracted by the first unit. The second unit makes contact with various services in order to gather the necessary information, which includes the following features:

URL length: The entire amount of characters in the URL is used to determine this attribute. Phishing websites frequently conceal their genuine identities or hazardous information by using abnormally long URLs. A URL’s length can serve as a reliable predictor of a website’s legitimacy or possible danger. Trusted websites frequently utilize short, straightforward URLs, but phishing sites typically employ longer, more complicated URLs to trick consumers.

Domain age: This feature shows how many days have passed since the domain was first registered. Newly registered domains are frequently used by phishing websites to evade detection by security measures. Because attackers often abandon domains rapidly, resulting in a cycle of registration and deactivation, a newly registered domain is frequently more suspect. Using programs like Python to query domain registration databases will yield this feature.

Presence of HTTPS: This feature determines whether the website communicates using HTTPS (Hypertext Transfer Protocol Secure). Phishing websites may not use HTTPS or may use it dishonestly with a phoney certificate, despite the fact that HTTPS guarantees encrypted interaction between the user and the website. Because some phishing sites employ HTTPS to look authentic, a missing or faulty HTTPS connection can, therefore, be a sign of a phishing website. However, this isn’t always the case.

Abnormal URL: Finding unusual patterns in the URL is how this characteristic is computed. Phishing websites sometimes disguise themselves by using techniques like misspelled words, extra character insertion, or domain name obfuscation. We can identify anomalies that point to phishing efforts by comparing the URL structure with well-known patterns of trustworthy websites. By including extra keywords or malicious redirection, this feature assists in identifying URLs that attempt to imitate well-known websites.

Special characters count: This feature keeps track of how many special characters (such as @, &, %, and #) are present in the URL. Phishing websites frequently use a lot of special characters to deceive automated detection systems or make the URL appear authentic. Since special characters are commonly used to obscure the genuine purpose of a website, a high number of them in the URL may be a sign of a phishing effort. This feature aids in identifying phishing efforts that employ techniques such as disguised links, excessive punctuation, or URL shortening.

The data is systematically fed into the trained classifier after all the characteristics have been acquired. The trained machine learning classifier then outputs its prediction as either “phishing” or “benign.” In order to prohibit phishing URLs, the client then retrieves this response. The missing feature would be given a value of -1 if any of the external parameters could not be found, such as a website that did not reply to PING requests or a missing domain age. The classifier was set up for testing on a cloud instance of Amazon Web Services (AWS).

A sample cloud server response to a benign URL and a phishing URL are displayed in Figs. 6 and 7, respectively. The server uses a straightforward RESTful call technique, as seen in Figs. 6 and 7, in which the URL to be examined is supplied to the server as an argument via the HTTP protocol. The reply is then sent in the plain text “phishing” or “benign.” To stop the user from clicking on phishing URLs, this response can be sent to a web browser plugin, email client plugin, or even a mobile text messaging plugin. On average, each URL took 11.5 μs to complete a request on the cloud.

Figure 6 Overview of server-side operation.

Figure 7 Sample phishing web page.

Result and discussions

The performance analysis of the proposed phishing detection method is shown in this section, with emphasis provided on important metrics, including accuracy, precision, recall, and F1-score. It highlights the efficacy of the proposed methodology and tackles phishing detection issues by contrasting the outcomes with those of current techniques. Within the framework of earlier studies, the discussion delves more into the approach’s advantages and possible enhancements. The parameter and their values are shown in Table 3.

Table 3 Parameters and their values.

Parameters	Values	
Epoch	200	
Learning rate	0.01	
Momentum	0.9	
Decay	10−5	
Batch size	64	
Dropout rate	0.5 to 0.1	

Experimental setup

Each test was carried out on a Windows 10 computer that had an Intel Core i5 processor running at 2.60 GHz and 16 GB of RAM installed. Using TensorFlow and Keras as the main deep learning libraries, the suggested framework was put into practice using Python in the Anaconda3 environment. With a batch size of 64 and an initial learning rate of 0.001, the SSOA optimizer was used to train the model. With a patience of 200 epochs, early halting was implemented based on validation loss to avoid overfitting. A maximum of 100 epochs were used for the training process. Each dataset (ISCX-URL-2016, URL-Based Phishing Dataset, Mendeley_2020, and PhishStorm) was split into 80% training and 20% testing sets in order to assess the suggested method’s capacity for generalization. All baseline and proposed models consistently used this stratified splitting technique to ensure equitable performance comparisons under consistent experimental settings.

Description of the dataset

The ISCX-URL-2016, URL-Based Phishing Dataset, Mendeley_2020, and Phishstorm datasets were chosen due to their realistic representations of benign and phishing URLs, diversity, and extensive feature sets. They are sizable enough to ensure the efficient training and testing of deep learning models, cover a variety of phishing categories, and incorporate a broad spectrum of URL attributes. Additionally, these datasets provide comprehensive details that are necessary for accurate phishing detection, like WHOIS information and domain age. ISCX-URL-2016: This dataset includes 114,400 URLs categorized as phishing, spam, malware, or vandalism. It contains features like URL length, special characters, and linguistic properties, aiding in the identification of phishing websites.

URL-Based Phishing dataset: From Kaggle, this dataset contains 11,054 records with 33 attributes, including URL structure and domain features. It helps distinguish between phishing and benign URLs.

Mendeley_2020: Divided into small (58,645 instances) and large (88,647 cases) subsets, this dataset includes 111 features per URL, including domain age, DNS records, and WHOIS information, useful for identifying phishing websites.

Phishstorm: This dataset contains 60,000 benign and 95,541 phishing URLs. It includes features like brand names, URL tokens, and HTTPS usage, offering insights into phishing tactics.

Evaluation criteria

Several efficiency metrics, including false positive rate (FPR), accuracy (A), precision (P), recall (R), and F1-score, were used to assess the efficacy of the suggested strategy.

The accuracy shows the rate at which the model correctly classifies the input URLs after training. The calculation involves dividing the total number of true positive (TP) and true negative (TN) instances by the overall amount of FP, FN, TN, and TP instances.

(23) Acc=Tp+TnTp+Tn+Fp+Fn

From the overall number of correctly predicted instances, precision provides the TP rate. By achieving high accuracy, it is feasible to show the recommended model’s great efficiency.

(24) Prec=TpFp+Tp

The recall is the total amount of TP overall expected samples. The performance of the proposed model is shown by a higher R-value close to one.

(25) Rec=TpTp+Fn

To show the expected sample proportion, the FPR employs a phishing URL. For a superior performance strategy, a low FPR close to zero is therefore necessary.

(26) FPR=FpTn+Fp

The F1-score shows the harmonic mean of P and R.

(27) F1=2×Prec×RecPrec+Rec

Matthew’s correlation coefficient (MCC) is utilized for datasets of varying class sizes and is considered to be balanced. MCC offers a correlation coefficient among expected and actual results.

(28) MCC=TP∗TN−FP∗FN(TP+FP)(TP+FN)(TN+FP)(TN+FN)

Figure 8 displays the distribution of the main features that were taken out of the dataset. The Histogram of Extracted Features reveals developments in the frequency of variables like URL length and special characters. This provides information about feature attributes that increase detection accuracy and aids in identifying criteria that differentiate benign URLs from phishing ones. It is easier to comprehend how these features affect the model’s performance due to the histogram.

Figure 8 Histogram of extracted features.

The balanced data representation is shown in Fig. 9.

Figure 9 Shows the outcome of the balanced dataset.

Using the data imbalance approach SPE-ACGAN approach, the minority samples present in our proposed data are balanced.

In order to qualitatively evaluate the selected URL variables, quantized DL variables for the primary phishing attack attributes that SOM reflects are displayed in Fig. 10. By assigning URLs to various sites based on each chosen rule, a qualitative depiction of the feasibility of the optimized collection of characteristics was produced.

Figure 10 Visualization of selected features.

The training and testing accuracy and loss of proposed datasets are shown in Fig. 11. We trained our proposed approach over 200 epochs. The learning rate was set as 0.01.

Figure 11 Graphical representation of training and testing accuracy and loss for the proposed approach.

The performance of different methods on the Mendeley_2020 dataset is displayed in Table 4, and the graphical representation is shown in Fig. 12. With 99.36% accuracy, 99.24% precision, 99.18% recall, 99.21% F1-score and 99.28% MCC, the suggested approach outperforms models such as Random Forest (97.15% accuracy) and DNN+LSTM (98.98% accuracy). In comparison to other models like LightGBM and Gradient Boost, it consistently produces better outcomes across all measures, demonstrating its efficacy.

Table 4 Contrasting the performance of models using the Mendeley_2020 dataset.

Approaches	Class	Pre	Rec	FS	MCC	Acc	
Random forest	Phishing	95.10%	96.50%	95.80%	96.52%	97.15%	
Legimate	96.46%	95.80%	96.00%	
Average	95.78%	96.13%	95.90%	
Bagging	Phishing	95.30%	94.50%	94.90%	95.55%	95.78%	
Legimate	96.30%	95.30%	95.60	
Average	95.80%	94.93%	95.33%	
GradientBoost	Phishing	92.50%	94.00%	93.25%	94.33%	95.37%	
Legimate	93.62%	93.16%	93.39%	
Average	93.06%	93.58%	93.32%	
LightGBM	Phishing	95.00%	95.50%	95.25%	95.92%	96.67%	
Legimate	95.50%	95.06%	95.11%	
Average	95.08%	95.28%	95.18%	
DNN	Phishing	91.00%	−	91.50%	91.26%	91.13%	
Legimate	91.78%	−	91.14%	
Average	91.39%	–	91.39%	
DNN+LSTM	Phishing		99.10%	98.95%	98.99%	98.98%	
Legimate	98.88%	99.04%	
Average	–	98.99%	99.01%	
RNN	Phishing	−	−	97.35%	97.26%	97.22%	
Legimate	−	−	97.25%	
Average	–	–	97.30%	
CNN	Phishing	−	−	93.20%	93.36%	93.33%	
Legimate	−	−	93.59%	
Average	–	–	93.39%	
Proposed	Phishing	99.20%	99.10%	99.15%	99.28%	99.36%	
Legimate	99.28%	99.26%	99.27%	
Average	99.24%	99.18%	99.21%	

Figure 12 Differentiation of Mendeley_2020 dataset.

The ISCX-URL dataset is used in Table 5 to compare how well various strategies work. With 99.18% accuracy, 99.06% prec, 99.01% rec, 99.03% FS and 99.10%MCC, the suggested approach performs better than any other.

Table 5 Contrasting the performance of models using the ISCX-URL dataset.

Approaches	Class	Precision	Recall	F1-score	MCC	Accuracy	
Auto encoder-DNN	Phishing	90.30%	91.50%	90.90%	92.11%	92.77%	
Legimate	89.70%	91.20%	91.00%	
Average	90%	91.36%	91.45%			
PDSMV3-DCRNN	Phishing	99.05%	99.20%	99.12%	99.05%	99.08%	
Legimate	98.90%	98.90%	98.95%	
Average	98.98%	99.05%	99.03%			
Variational auto encoder-DNN	Phishing	97.10%	97.70%	97.40%	97.67%	97.89%	
Legimate	97.30%	97.40%	97.50%	
Average	97.20%	97.54%	97.45%			
Sparse auto encoder-DNN	Phishing	93.60%	94.10%	93.85%	95.34%	95.83%	
Legimate	94.00%	94.50%	94.30%	
Average	93.82%	94.81%	94.85%			
Proposed	Phishing	99.10%	99.15%	99.12%	99.10%	99.18%	
Legimate	99.02%	98.90%	99.00%	
Average	99.06%	99.01%	99.03%			

It outperforms models such as Sparse AE-DNN and AE-DNN and produces better results than PDSMV3-DCRNN (accuracy of 99.08%) and VAE-DNN (accuracy of 97.89%). This indicates that the proposed strategy offers the best accuracy and a well-balanced performance in terms of F1-score, precision, and recall. Figure 13 shows the differentiation of various methods using the ISCX-URL dataset.

Figure 13 Differentiation of ISCX-URL dataset.

The URL dataset is used in Table 6 to compare the effectiveness of different strategies. The proposed approach achieves the best results across all parameters, outperforming all current models with 99.12% accuracy, 99.04% precision, 99.02% recall, 99.03% F1-score and 99.08% MCC. While LR+SVC+DT (98.12% accuracy) and RF (96.77% accuracy) perform well, they are still inferior to the proposed approach. The proposed method is more effective in terms of precision and recall than SVM, which performs lowest with an accuracy of 71.8%. Figure 14 illustrates the distinction of various methods employing the URL dataset.

Table 6 Contrasting the performance of various methods using URL dataset.

Approach	Class	Prec	Rec	FS	MCC	Acc	
DT	Phishing	95.5%	96.4%	95.80%	95.66%	95.41%	
Legimate	96.1%	95.6%	96.02%	
Average	95.8%	96%	95.91%		
LR+SVC+DT	Phishing	97.60%	95.80%	95.90%	96.98%	98.12%	
Legimate	97.02%	96.86%	95.88%	
Average	97.31%	96.33%	95.89%			
SVM	Phishing	96.80%	48.00%	65.90%	68.60%	71.8%	
Legimate	95.88%	51.62%	65.44%	
Average	96.34%	49.81%	65.67%			
RF	Phishing	96.60%	97.90%	97.20%	96.94%	96.77%	
Legimate	96.86%	97.12%	97.04%	
Average	96.73%	97.51%	97.12%			
Proposed	Phishing	99.00%	99.10%	99.04%	99.08%	99.12%	
Legimate	99.08%	98.94%	99.02%	
Average	99.04%	99.02%	99.03%			

Figure 14 Differentiation of URL dataset.

The performance of different methods on the Phishstorm dataset is contrasted in Table 7. According to all criteria, the proposed strategy performs best, achieving 99.17% accuracy, 99.10% recall, 99.06% precision, 99.08% F1-score and 99.12% MCC. With 97.10% and 95.05% accuracy, respectively, Texception Net and GA perform well in comparison, although they are still not as good as the proposed strategy. LSTM and CNN perform worse, especially in recall and precision. The performance of the Phishstorm dataset is shown in Fig. 15.

Table 7 Comparing proposed approaches with current methods utilizing the Phishstorm dataset.

Methods	Class	Recall	Precision	F-score	MCC	Accuracy	
CNN	Phishing	85.00%	86.80%	–	87.85%	90.16%	
Legimate	83.24%	84.50%	–	
Average	84.12%	85.65%	–		
LSTM	Phishing	–	85.10%	–	86.06%	87.77%	
Legimate	–	83.70%	–	
Average	–	84.40%	–		
URLNet	Phishing	–	89.40%	88.74%	90.81%	93.95%	
Legimate	–	87.88%	87.03%	
Average	–	88.64%	87.89%		
CNN-LSTM	Phishing	87.30%	88.60%	87.94%	90%	92.29%	
Legitimate	86.50%	87.10%	86.80%	
Average	–	87.85%	–		
Texception net	Phishing	88.90%	92.80%	90.81%	93.52%	97.10%	
Legitimate	87.40%	91.74%	89.61%	
Average	88.15%	92.27%	90.21%		
GA	Phishing	90.40%	94.10%	92.20%	93.35%	95.05%	
Legitimate	89.84%	92.54%	91.24%	
Average	90.12%	93.32%	91.72%		
Proposed	Phishing	99.10%	99.20%	99.15%	99.12%	99.17%	
Legitimate	99.02%	99.00%	99.01%	
Average	99.06%	99.10%	99.08%		

Figure 15 Performance differentiation of various approaches using the Phishstorm dataset.

The performance of different optimization techniques is compared in Table 8 and Fig. 16, which demonstrates that the suggested optimizer performs better than the others across the board. In comparison to algorithms like Adam, Adamax, and SGD, it achieves the highest accuracy (99.28%), precision (99.16%), recall (99.24%), and F1-score (99.20%), demonstrating a greater capacity to create accurate, exact, and dependable predictions. This indicates that when it comes to handling the optimization process, the proposed optimizer provides superior overall performance.

Table 8 Differentiation of proposed with prior optimization methods.

Optimizer	Accuracy	Precision	Recall	F1-score	
Adam	98.89%	98.95%	98.79%	98.87%	
Adagrad	58.14%	75.52%	55.28%	62.10%	
Adamax	98.22%	98.44%	98%	98.22%	
SGD	82.65%	86.30%	80.14%	82.59%	
Nadam	98.81%	98.85%	98.66%	98.75%	
RMSprop	98.27%	98.35%	98.12%	98.24%	
Proposed	99.28%	99.16%	99.24%	99.20%	

Figure 16 Comparison of various optimization approaches.

Figure 17 shows distinctions both with and without the feature selection approach. The suggested feature selection method performs better when compared to current feature selection techniques.

Figure 17 Distinguishing between the proposed and current methods, both including and excluding the feature selection approach.

Table 9 presents the performance of the proposed feature selection (RKOA) and optimization (SSOA) methods across four phishing detection datasets. The results demonstrate consistently high accuracy ranging from 99.12% to 99.36%, with low computational time between 9.01 ms and 13.25 ms. This highlights the effectiveness and efficiency of the proposed approach in selecting relevant features and optimizing the model.

Table 9 Feature selection and optimization performance across our proposed datasets.

Proposed datasets	Feature selection method	Optimization approach	Accuracy	Time (ms)	
ISCX-URL-2016	RKOA	SSOA	99.18%	12.05	
URL based Phishing	RKOA	SSOA	99.12%	13.25	
Mendeley-2020	RKOA	SSOA	99.36%	9.01	
Phishstorm	RKOA	SSOA	99.17%	10.11	

The suggested feature extraction method is compared with the current feature extraction methods, including CNN, LSTM, CNN-LSTM, RNN, and CNN-BiLSTM. Figure 18 illustrates how the suggested model outperforms current models while producing better results.

Figure 18 Comparison of suggested and current extraction of features methods.

Table 10 and Fig. 19 contrast the effectiveness of several strategies with the suggested technique in terms of the following essential metrics: F1-score, accuracy, precision, recall, MCC and false positive rate (FPR). With a low FPR (0.34%) and the maximum accuracy (99.72%), precision (99.76%), F1-score (99.54%) and MCC (99.58%), the suggested method is a very effective and dependable model. With a much lower FPR, this model offers a better balance than other models such as CNN Attention (99.31%) and faster region-based convolutional neural network (Faster RCNN) (99.68%), outperforming them in precision and recall as well as accuracy and F1-score. Models such as RNN+CNN and Bayes Net exhibit reduced accuracy and precision, underscoring the efficacy of the proposed strategy.

Table 10 Overall performance evaluation.

Methods	Acc	Prec	Rec	FS	FPR	MCC	
DNN	99.52%	–	–	–	–	–	
LSTM	99.57%	–	–	–	–	–	
CNN	99.43%	97%	98.2%	97.6%	–	98.51%	
Bayes Net	92.8%	93.4%	90.1%	91.7%	–	92.24%	
Faster RCNN	99.68%	99.13%	88.67%	–	–	–	
RNN+CNN	95.79%	94.24%	97.27%	95.73%	–	95.76%	
CNN attention	97.82%	99.74%	98.89%	99.31%	0.63%	98.56%	
Proposed	99.72%	99.76%	99.32%	99.54%	0.34%	99.58%	

Figure 19 Overall differentiation of proposed with existing approaches.

Table 11 highlights the training and inference times of the proposed model across different phishing datasets. Training time reflects how long the model takes to learn from each dataset, while inference time indicates the speed at which it predicts outcomes per sample. The results show that the model maintains low inference times under 1.1 ms, demonstrating its suitability for real-time phishing detection.

Table 11 Comparison of Training time and Inference time.

Dataset	Training time (s)	Inference time (ms)	
ISCX-URL-2016	125.4	1.02	
URL-based phishing	196.8	0.96	
Mendeley-2020	194.2	0.89	
PhishStorm	202.6	0.93	

Ablation study

We performed ablation research by gradually deleting or replacing essential components in order to assess the contributions of each component in our phishing detection framework. The impact of each element on overall performance, including accuracy, precision, recall, and F1-score, is investigated in this study.

The incremental impact of different phishing detection pipeline components across four distinct datasets is shown in Table 12. The more steps that are added to the process, the more accurate it becomes. First, preprocessing produces high accuracy in every dataset. Performance is significantly improved by adding data imbalance treatment, underscoring the need to resolve class imbalances. Accuracy in feature extraction and selection keeps increasing, highlighting how important it is to choose the most pertinent features for phishing detection. Lastly, the most significant improvement and best accuracy are obtained with the addition of classification and URL encoding. This demonstrates how every stage helps to improve the model’s capacity to recognize phishing URLs with accuracy.

Table 12 Comparison of ablation study based on the accuracy.

Methods	Dataset 1	Dataset 2	Dataset 3	Dataset 4	
Preprocessing	98.72%	98.83%	99.01%	98.56%	
Preprocessing + data imbalance	98.96%	98.95%	99.03%	98.72%	
Preprocessing + data imbalance + Feature extraction	99.04%	98.99%	99.08%	98.99%	
Preprocessing + data imbalance + Feature Extraction + Feature selection	99.12%	99.08%	99.15%	99.07%	
Preprocessing + data imbalance + Feature Extraction + Feature selection + Classification	99.21%	99.12%	99.18%	99.24%	
Preprocessing + data imbalance + Feature Extraction + Feature selection + Classification URL encoding	99.45%	99.28%	99.30%	99.28%	

Discussions

A crucial cybersecurity tactic is phishing detection, which aims to spot and stop fraudulent efforts to steal private data by impersonating trustworthy organizations. As phishing websites and emails have become more prevalent, sophisticated detection methods like machine learning and deep learning are being used to protect users from economic and reputational damage.

Existing limitations

While applicable in certain situations, current phishing detection techniques have a number of significant drawbacks. The accuracy of phishing detection is increased by hybrid models such as DNN-LSTM, which combine character-based connections with high-level NLP characteristics. However, they are not appropriate for real-time applications due to their computational complexity. Similar to this, CNN-based methods have limited scalability since they rely on particular kernel variants, which limits their capacity to adapt to a variety of datasets despite being lightweight and successful in feature selection. Although they have domain-specific advantages, novel approaches like the binary bat algorithm and LBPS with BP neural networks struggle with generalizability and broader applicability. Moreover, hybrid feature-based techniques such as PDHF include delays that are inappropriate for time-sensitive situations while optimizing accuracy in the order of computing overhead.

Overcoming limitations with the proposed method

The novel two-stage methodology of the proposed OptSHQCNN architecture successfully addresses these issues. The robustness of feature selection is increased during the pre-deployment phase by thorough preprocessing, which removes invalid and unnecessary features, and the CBAM, which isolates crucial properties. The complexity seen in hybrid models is reduced using the RKOA, which guarantees a computationally effective selection of essential features. In order to improve scalability and applicability across a variety of datasets, OptBERT’s integration for URL encoding after deployment guarantees that the model preserves significant contextual information. By further optimizing classification and lowering computational costs, SHQCNN can be fine-tuned using SSOA to improve real-time performance.

A thorough validation process was carried out using four different benchmark datasets: ISCX-URL-2016, URL-Based Phishing Dataset, Mendeley_2020, and PhishStorm in order to guarantee the stability and generalizability of the proposed OptSHQCNN model. We are able to show the adaptability of our feature selection and optimization techniques in a variety of data settings because these datasets vary in terms of structure, source, and distribution. RKOA and SSOA integration maintained stability in classification metrics across datasets while also ensuring effective feature reduction. The framework also exhibits great potential for real-time implementation in browser-based and edge-security applications by reducing computing overhead and utilizing effective encoding with OptBERT, making it scalable to massive data quantities.

Benefits of the proposed method

Compared to current approaches, the OptSHQCNN framework has the following benefits: High accuracy and efficiency: Over 99% accuracy, precision, recall, and F1-score are attained by combining sophisticated feature extraction and selection approaches.

Reduced computational complexity: The model’s operations are streamlined by optimization methods such as RKOA and SSOA, which qualifies it for real-time detection scenarios.

Scalability and adaptability: The model may generalize over a variety of datasets by utilizing OptBERT for URL encoding, which overcomes the drawbacks of domain-specific methods.

Effective feature selection: By ensuring that only the most pertinent features are taken into consideration, the CBAM model lowers noise and improves classification results.

Broad applicability: The two-stage framework can be expanded to a number of real-world uses, including personal terminals, network edges, and browser plugins.

Proposed limitations

Our proposed method performs better than current approaches and produces outstanding outcomes, according to testing and evaluation. However, the suggested system does have several drawback. The model does not determine whether the website is live by looking at its URL, which affects the findings. In order to get around this restriction, we might need to enhance feature engineering and speed up the training process. This would enable us to confirm the website’s current status and increase the correctness of the training process.

Conclusion and future scope

In order to tackle the crucial problem of phishing detection, the proposed investigation presents OptSHQCNN, an effective and novel two-stage deep learning architecture. In order to ensure thorough data handling and feature analysis, the methodology is divided into pre-deployment and post-deployment phases. Data imbalances, irrelevant features, and missing values are efficiently managed by pre-deployment preprocessing, and the convolutional block attention module (CBAM) precisely improves feature extraction. The red kite optimization algorithm (RKOA) increases model efficiency by making sure that only essential features are chosen. By fine-tuning hyperparameters with the shuffled shepherd optimization algorithm (SSOA), the shallow hybrid quantum-classical convolutional neural network (SHQCNN) improves classification accuracy even more. Post-deployment URL encoding using Optimized Bidirectional Encoder Representations from Transformers (OptBERT) enhances feature representation and classification by capturing the complex patterns of phishing URLs.

The results, which showed over 99.72% accuracy, 99.76% precision, 99.32% recall, 0.34% FPR and 99.54% F1-score, demonstrate the effectiveness that the proposed structure performs in comparison to current techniques. Numerous benefits arise from the model’s capacity to incorporate advanced deep learning and optimization methodologies. It provides great dependability by reducing false positives and negatives. Scalability and adaptation to various datasets and phishing scenarios are made feasible by the modular two-stage architecture. OptBERT’s contextual encoding ensures improved identification of intricate phishing URLs, while quantum-inspired SHQCNN speeds up computations without impacting accuracy. These additions enable the framework to be used in real-world applications like browser plugins and instant messaging app integrations, promoting safer online spaces and serving as a standard for phishing detection research in the future. While this study emphasizes URL-based feature extraction due to its suitability for real-time detection, it acknowledges the limited consideration of other informative aspects, such as the visual similarity of web content and user interaction behaviour. Incorporating these features in future work may further enhance the model’s performance in complex and deceptive phishing scenarios.

We promulgate that this manuscript is authentic, has not been divulged before, and is not currently being contemplated for publication otherwhere.

Additional Information and Declarations

Competing Interests

The authors declare that they have no competing interests.

Author Contributions

Srikanth Meda conceived and designed the experiments, performed the experiments, analyzed the data, performed the computation work, authored or reviewed drafts of the article, and approved the final draft.

Vangipuram Sesha Srinivas conceived and designed the experiments, performed the experiments, analyzed the data, performed the computation work, authored or reviewed drafts of the article, and approved the final draft.

Killi Chandra Bhushana Rao conceived and designed the experiments, performed the experiments, analyzed the data, performed the computation work, prepared figures and/or tables, and approved the final draft.

Repudi Ramesh conceived and designed the experiments, performed the experiments, analyzed the data, performed the computation work, prepared figures and/or tables, authored or reviewed drafts of the article, and approved the final draft.

Narasimha Rao Yamarthi conceived and designed the experiments, performed the experiments, analyzed the data, performed the computation work, prepared figures and/or tables, authored or reviewed drafts of the article, and approved the final draft.

Data Availability

The following information was supplied regarding data availability:

The data is available at Figshare.

Yamarthi, Dr. Narasimha Rao (2025). Advanced Phishing Detection. figshare. Dataset. https://doi.org/10.6084/m9.figshare.28130225.v1.

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
