# Peer review of "A dual-phase deep learning framework for advanced phishing detection using the novel OptSHQCNN approach"

_PeerJ Computer Science, doi:10.7717/peerj-cs.3014_

## Round 0.1 · original submission · Major Revisions

Please address all reviewer comments.

Reviewer 1 ·

Basic reporting

The manuscript explores an ambitious combination of advanced techniques, but it’s not entirely clear how all these components come together. The paper would benefit from a clearer explanation of each technology’s role and its specific contribution to the overall framework. Right now, the connections between them feel a bit disjointed, which makes it hard to follow the logic behind the approach.

The provided diagram is meant to clarify the methodology, but instead, it adds to the confusion. It’s difficult to understand how each part interacts, and a more structured, easy-to-follow visual representation would be really helpful.

One of the more surprising choices in the paper is the use of a hybrid quantum-CNN model for URL data. Typically, quantum networks are used in cases where quantum relationships exist in the data, so this approach seems unconventional. Given the details in the experimental setup, it’s also unclear whether a true quantum simulation was actually performed. Some additional explanation on why this approach was chosen and how it fits into the problem at hand would strengthen the argument.

The manuscript also discusses using Generative Adversarial Networks (GANs) for data balancing, but it’s not clear how these are implemented. Important details—like the specific GAN architecture, training process, and evaluation metrics—are missing. Providing these details would make it easier to assess the effectiveness and reproducibility of the approach.

Another issue is with the reporting of precision. It’s only provided for one class, but the paper doesn’t specify which class that is. Including a full set of evaluation metrics (such as precision, recall, and F1-score for all classes) would make the results much more informative and balanced.

The use of an Auxiliary Classifier GAN (ACGAN) for rebalancing a dataset composed mostly of URLs seems like overkill. There are simpler and more efficient ways to handle data rebalancing, such as basic oversampling techniques or less complex GAN architectures.

Hyperparameter optimization is mentioned, but it’s unclear what constraints were used, how the optimizer was configured, or what encoding strategies were applied. Without these details, it’s difficult to gauge the rigor and reliability of the optimization process.

Experimental design

Insufficient data is provided to validate the reported results. The experimental setup is unclear and needs to be significantly expanded. The role of the components discussed in the methodology in the conducted simulations should be clearly outlined to help readers understand their specific contributions. Furthermore, the overall experimental design lacks clarity, making it difficult to determine how the experiments were structured and whether they adequately support the conclusions drawn.

Validity of the findings

Insufficient data is provided to validate the reported results.
The experimental setup is unclear and needs to be significantly reworked.
The role of the components discussed in the methodology in the conducted simulations should be clearly discussed.

Reviewer 2 ·

Basic reporting

1. The study mainly focuses on feature extraction based on URLs, while the exploration of other potential features of phishing websites (such as the visual similarity of web content and user interaction behaviors) is relatively limited. It is suggested to further explore these features to improve the detection performance of the model in complex situations.
2. Some of the cited references are relatively old and do not fully reflect the latest research progress in this field. It is recommended to further update and improve the literature review section by adding citations and analyses of high-quality studies published in recent years, to make the literature review more up-to-date and comprehensive. The manuscript could be improved by citing more recent literature about the application of machine learning.

Experimental design

1. The training and inference times of the model on different datasets were not mentioned in the experiments, which is crucial for evaluating the model's practical application value. It is suggested to supplement this information to more comprehensively assess the model's performance.
2. Although some key parameters have been listed in the paper, the description of the specific details of some experiments (such as the specific process of model training) is still not detailed enough. It is suggested to supplement and improve these experimental details and provide a complete code implementation (if possible), so that other researchers can replicate the experimental results using the same method and verify the validity of the study.

Validity of the findings

Regarding feature selection and model optimization, it is recommended to further explore their applicability and stability across different datasets, as well as their scalability and real-time performance in practical application scenarios.

Additional comments

The explanations of some terms and concepts are not detailed enough, which may make it difficult for non-specialist readers to understand. It is suggested to further optimize the language expression, add explanations for key terms and concepts, simplify the structure of complex sentences, and make the paper more accessible.

---

## Round 0.2 · Minor Revisions

The authors correctly addressed the requests of the reviewers, but some changes are still needed. In particular, the binary classification results should be measured through the Matthews correlation coefficient (MCC) as well.

I invite the authors to add these results and their comments to a new version of the manuscript.

Reviewer 1 ·

Basic reporting

The authors have addressed my concerns.

Experimental design

Additional details have been added to the manuscript addressing my repeatability concerns.

Validity of the findings

Additional details have been added to the manuscript addressing my repeatability concerns.

---

## Round 0.3 · accepted · Accept

The authors correctly addressed my requests and therefore I can recommend this article for acceptance and publication.